# Green Preparation of Aminated Magnetic PMMA Microspheres via EB Irradiation and Its Highly Efficient Uptake of Ce(III)

**DOI:** 10.3390/ma15196553

**Published:** 2022-09-21

**Authors:** Yuan Zhao, Tian Liang, Pengpai Miao, Tao Chen, Xiaobing Han, Guowen Hu, Jie Gao

**Affiliations:** Hubei Key Laboratory of Radiation Chemistry and Functional Materials, Hubei University of Science and Technology, Xianning 437100, China

**Keywords:** magnetic PMMA microspheres, PEI, radiation induced grafting, Ce(III) adsorption

## Abstract

The modification of polymers can significantly improve the ability to remove rare earth ions from wastewater, but so far few studies have focused on the irradiation-induced grafting method. In this study, a novel magnetic chelating resin for Ce(III) uptake was first synthesized by suspension polymerization of PMMA@Fe_3_O_4_ microspheres followed by irradiation-induced grafting of glycidyl methacrylate (GMA) and subsequent amination with polyethyleneimine (PEI). The FT-IR, SEM, TG and XRD characterization confirmed that we had successfully fabricated magnetic PMMA-PGMA-PEI microspheres with a well-defined structure and good thermal stability. The obtained adsorbent exhibited a satisfactory uptake capacity of 189.81 mg/g for Ce(III) at 318.15 K and an initial pH = 6.0. Additionally, the impact of the absorbed dose and GMA monomer concentration, pH, adsorbent dosage, contact time and initial concentration were thoroughly examined. The pseudo-second order and Langmuir models were able to describe the kinetics and isotherms of the adsorption process well. In addition, the thermodynamic data indicated that the uptake process was spontaneous and endothermic. Altogether, this research enriched the Ce(III) trapping agent and provided a new method for the removal rare earth pollutants.

## 1. Introduction

Rare earth metals enjoy the reputation of being “industrial gold”, “industrial vitamins” and “military metals” in the military, the metallurgical industry, petrochemicals, glass ceramics, new materials and in other fields [1,2]. They are extremely important and non-renewable strategic resources [3]. As a strategic investment in high-tech industries, rare earth elements have a very wide range of applications [4]. Rare earth metals—due to their excellent magnetic, optical and catalytic properties—are widely used in the chemical, electrical and metallurgical fields [5,6]. Due to high demand, there are still lots of rare earth ions in the chemical precipitations of the wastewater produced by extensively mined rare earth deposits that cannot completely be recycled—resulting in a large number of rare earth ions remaining in the environment [7,8]. Therefore, it is of particularly great importance to develop a method that can quickly recover and separate rare earth ions.

At present, the recovery of rare earth ions mainly includes chemical precipitation, solvent extraction, membrane separation, ion exchange, etc. [9,10,11]. However, these methods commonly come with the problems of being high cost, easily causing secondary pollution in the environment and being unsuitable for industrial production [12,13]. Among these methods, adsorption is a preferable method for the recovery of rare earth ions due to its simplicity, low cost and effectiveness [14,15,16,17]. It is widely used in the recovery of low-concentration rare earth ions and is considered to be a promising rare earth recovery method [18,19].

In recent years, lots of adsorbents for rare earth ion recovery have been developed, such as biopolymeric-LDH hybrid nanocomposites used for the recovery of Ce^3+^ and La^3 +^ [20] and amino-functionalized magnetic graphene oxide for heavy REE adsorption [21]; LDH-intercalated cellulose nanocomposites can achieve adsorption equilibrium in 10 min and their adsorption capacity reaches 96.25 mg/g [22]. However, these nanocomposite adsorbents are costly and their preparation method is complicated. In contrast, polymer microspheres have become an ideal material for the recovery of rare earth ions due to their excellent properties [23].

Methyl methacrylate (PMMA) is a typical representative of polymer microspheres [24]. Generally, the challenge of PMMA being used in adsorption is that a more complicated solid/liquid separation is required at the end of the adsorption [25]. However, the use of magnetic matrix composite materials can eliminate this shortcoming [26,27]. Magnetic polymer microspheres have not only been widely used to separate materials and have served as carriers, but also have good performance in adsorption after modification [28]. As magnetic PMMA microspheres are easy to recycle after adsorption is completed, magnetic recovery technology is an important strategy for designing new adsorbents for rare earth ion recovery.

In order to make PMMA exhibit excellent properties, such as good thermostability, high selectivity and anti-interference, grafting modification is a good choice. Nowadays, irradiation-induced grafting of polymers is considered one of the most promising methods for improving the adsorption capacity of polymers. Radiation grafting technology has the advantages of its simple operation, easy control and there being no limitations in the substrates and monomers used [29]. Nowadays, it is widely used in the preparation of adsorption materials [30]. An adsorbent prepared by radiation grafting technology has functional groups mainly distributed on the surface of the substrate, with low monomer consumption, low cost, fast adsorption and desorption speeds and high utilization of functional groups as well as good selectivity [31]. GMA is a functional monomer containing both double bonds and epoxy groups; it can be easily grafted onto PMMA microspheres via electron beam irradiation.

In this work, we report a novel magnetic PMMA microsphere containing lots of amino groups for Ce(III) uptake. Firstly, magnetic PMMA microspheres were prepared by the suspension polymerization method. Secondly, GMA was grafted onto the magnetic PMMA microspheres through the electron beam co-irradiation method. Lastly, amination with PEI was used to obtain the magnetic PMMA-PGMA-PEI microspheres. The magnetic microspheres were characterized by SEM, XRD, FT-IR and TG. The influence of the radiation dose, pH, initial concentration, contact time and temperature were discussed. The adsorption kinetics, isotherms and thermodynamics of Ce(III) ions on magnetic PMMA-PGMA-PEI microspheres were systematically investigated. Lastly, the adsorption performance of the Ce(III) was studied and a possible uptake mechanism was proposed. Up to now, no research has focused on preparing magnetic PMMA-PGMA-PEI microspheres through the irradiation-induced grafting method. This research could provide a green and promising method for enhancing the removal of Ce(III) from wastewater.

## 2. Experimental

### 2.1. Materials and Chemicals

Nano Magnetite (Fe_3_O_4_, 99.5%, 20 nm, spherical), polyethyleneimine (PEI) and glycidyl methacrylate (GMA) were purchased from Aladdin Reagent (Shanghai, China). Oleic acid was supplied by Luoyang Chemical Reagent Co., Ltd. (Luoyang, China). Chlorophosphonazo, methyl methacrylate, sodium hydroxide, petroleum ether, ethanol, benzoyl peroxide and cerium nitrate hexahydrate were purchased from Sinopharm Chemical Reagent Co., Ltd. (Shanghai, China). Magnesium chloride and paraffin were supported by Tianjin Beichen founder Chemical Reagent Co., Ltd. (Tianjin, China). All chemicals were of analytical reagent grade and used as received. Deionized distilled water (18.25 MΩ) was used throughout the whole study for solution preparation.

### 2.2. Synthesis of Magnetic PMMA-PGMA-PEI

The route of magnetic PMMA-PGMA-PEI microspheres is schematically illustrated in Figure 1, including four steps: (1) Modification of Fe_3_O_4_ nanoparticles; (2) Synthesis of magnetic PMMA microspheres; (3) Grafting of magnetic PMMA microspheres with GMA; (4) Amination of magnetic PMMA-PGMA.

The first process for Fe_3_O_4_ modification is as follows: In detail, 4.0 g Fe_3_O_4_ nanoparticles and 4.0 g oleic acid added into 100 mL alcohol were kept in a 250 mL three-neck flask, stirring the mixture for 3 h at 80 °C in a water bath. Then, the modified Fe_3_O_4_ nanoparticles were separated and washed with distilled water and anhydrous ethanol several times.

The magnetic PMMA microspheres were prepared according to the following steps. Firstly, 0.8 g sodium hydroxide and 2.0 g magnesium chloride were dissolved with 80 mL distilled water under stirring at 333.15 K; the above mixed solution was placed in a 250 mL three-mouth round-bottom flask. Then, 40 g MMA, 4 g divinylbenzene, 0.3 g benzoyl peroxide, 0.2 g modified Fe_3_O_4_ and 6 mL paraffin was added with high-speed stirring at 343.15 K for 2 h. The product was collected by vacuum filtration and washed with distilled water and ethanol three times, respectively. After filtration, the obtained product was dried in an oven at 323.15 K for 10 h. The pure PMMA microsphere was prepared similarly to the above method, except without the addition of the Fe_3_O_4_ nanoparticles.

In the third step, GMA was grafted onto the surface of the magnetic PMMA through electron beam irradiation. In a typical preparation process, 3.0 g of magnetic PMMA microspheres were accurately weighed and packaged into a plastic bag and the bag was vacuum sealed. Then, 20 mL of 30% (*v*/*v*) glycidyl methacrylate methanol solution (GMA) solution was injected into the bag with a syringe. The sealed samples were irradiated by a 1 MeV electron accelerator (Wasik Associates, Dracut, MA, USA) from 20 to 120 kGy, with a dose rate of 20 kGy/pass. Then, the samples were filtered, washed with methanol and water to remove the monomers and byproducts and dried at 323.15 K for further use. The grafting yield (GY) was calculated using Equation (1) [30]:(1)$$GY(\%)=\frac{W_{g}-W_{0}}{W_{0}}\times 100\%$$
where *W*_0_ (g) and *W_g_* (g) are the weights of the magnetic PMMA and magnetic PMMA-PGMA, respectively.

Amination reaction: in a typical procedure, 1.0 g of the above GMA grafted microspheres were suspended in 10 mL C_2_H_5_OH and 5 g polyethyleneimine (PEI) was added. The above mixed solution was placed in a 50 mL round-bottom flask at 70 °C with stirring to induce a reflux reaction for 10 h, filtered and washed with ethanol and deionized water for several times and dried in an oven at 50 °C for 10 h. The accurate weight and quality of the magnetic PMMA-PGMA-PEI microspheres were recorded.

### 2.3. Characterization

The FT-IR spectra were recorded by a NICOLET 5700 spectrometer (Thermo Fisher Nicolet, Waltham, MA, USA). The crystal structures of the microspheres were investigated by X-ray powder diffraction (LabX XRD-6100, Shimazdu, Japan). The surface morphology was observed by scanning electron microscopy (VEGA-3 SBH, Tescan, Czech Republic). The thermal stability of the microspheres samples was characterized on a NETZSCH thermogravimetric analyzer (TG 209F3, Germany).

### 2.4. Bath Adsorption Experiments of Ce(III)

The Ce(III) adsorption performance of the magnetic PMMA-PGMA-PEI was evaluated via batch tests in triplicate. A series of flasks, each containing 20 mL Ce(III) solution at a specific concentration with a certain amount magnetic PMMA-PGMA-PEI, were agitated on an incubator shaker at a rate of 180 rpm for a given temperature and time.

The Ce(III) solutions remained at pH 6.0 in all the adsorption tests except for the test of pH on adsorption. The pH effect was investigated at an initial concentration of 200 mg/L in the pH range of 2 to 7; strong alkaline solution were not studied to avoid the formation of Ce(III) hydroxides. In this research, 0.1 M HCl or 0.1 M NaOH solution were used to adjust the pH values.

The effect of contact time was studied at an initial Ce(III) concentration of 200 mg/L at pH = 6. After each adsorption test, the filtrate was collected by membrane separation using a 0.45 μm polypropylene syringe filter. The Ce(III) concentrations of the filtrates were determined by chlorophosphonazo at a wavelength of 668 nm. The experiment on adsorption kinetics was conducted at predetermined time intervals (0–450 min) and the residual Ce(III) was determined by a UV-vis spectrophotometer (UV1901).

All the uptake experiments were conducted in triplicate, and the uptake capacities (mg/g) and removal efficiency (R, %) were calculated by [32,33,34]:(2)$$Q=\frac{(C_{0}-C_{t})V}{m}$$
(3)$$R=\frac{C_{0}-C_{t}}{C_{0}}\times 100\%$$
where *C*_0_ and *C_t_* are the initial and equilibrium concentrations of the Ce(III) ions, respectively, while *m* (mg) and *v* (mL) represent the adsorbent weight and the Ce(III) solution volume, respectively.

## 3. Results and Discussion

### 3.1. FT-IR

Figure 1 displays the FTIR spectra of the PEI, magnetic PMMA, magnetic PMMA-PGMA and PMMA-PGMA-PEI. For the Fe_3_O_4_ nanoparticles, an Fe-O stretching vibration was usually present at 581 cm^−1^ [35]. For the PEI, the absorption band included at 3284 cm^−1^ was attributed to a N-H stretching vibration. In the magnetic PMMA microspheres’ spectrum, a new band representing carbonyl stretching appeared at 1724 cm^−1^ [36]. Besides this, bands of C-H stretching vibrations at 2821 cm^−1^ and C-H bending vibrations at 1375 cm^−1^ and 1463 cm^−1^ appeared. Compared to the magnetic PMMA microspheres’ spectrum, the characteristic absorption peaks of the magnetic PMMA-PGMA were the epoxy vibration absorption at 996 cm^−1^ and the band at 1724 cm^−1^ (O-C=O stretching), which became stronger in intensity—providing evidence for the graft copolymerization of GMA on the magnetic PMMA microspheres. The disappearance of the epoxy vibration peaks and the N-H vibration absorption at 1598 cm^−1^ confirmed that the magnetic PMMA-PGMA microspheres were successfully modified with PEI [37].

### 3.2. XRD Analysis

The XRD patterns for the Fe_3_O_4_ nanoparticles, PMMA, magnetic PMMA, magnetic PMMA-PGMA and magnetic PMMA-PGMA-PEI are presented in Figure 2. As observed in the curve for the Fe_3_O_4_, strong peaks appeared at 2θ = 30.14, 35.41, 43.17, 53.44, 57.09 and 62.75°, which respectively correspond to the cubic spinel Fe_3_O_4_ crystal planes of (220), (311), (400), (422), (511) and (440). In addition, these peaks were consistent with JCPDS card, no.75–1610, as reported in the previous report [27]. Besides this, the typical diffraction peaks of the PMMA exhibited broad peaks at approximately 15.04°, which represented features of amorphous substances. For magnetic PMMA, magnetic PMMA-PGMA and PMMA-PGMA-PEI, the characteristic peaks of the Fe_3_O_4_ still existed—indicating the existence of Fe_3_O_4_ on the magnetic PMMA-PGMA-PEI.

### 3.3. Morphology Analysis

The morphology and appearance of the magnetic PMMA, magnetic PMMA-PGMA and magnetic PMMA-PGMA-PEI microspheres are represented in Figure 3; the magnetic PMMA microspheres had a high sphericity, with a particle size of about 500 μm and a smooth surface [38]. Compared with the magnetic PMMA microspheres, the particle size of the magnetic PMMA-PGMA microspheres was obviously increased—ranging from 500–550 μm (Figure 3c,d). This can be ascribed to swelling and the grafting of a large amount of PGMA during the modification. In addition, the surface of the magnetic PMMA microspheres changed from smooth to rough, which further proved the successful grafting and modification of the magnetic PMMA microspheres. Meanwhile, after the reaction with PEI, the surface of the magnetic PMMA-PGMA-PEI microspheres became rougher. However, the particle size was not obviously increased. The microscopic changes of the morphology can also be reflected by the BJH test results (Appendix A). The BET surface area, pore volume and average pore diameter of Magnetic PMMA-PGMA-PEI were lower than that of pristine Magnetic PMMA. Because the grafting of organic functional groups occupied some spaces of the pores of Magnetic PMMA after chemical modification, resulting in reduced specific surface area, pore volume and average pore diameter of Magnetic PMMA-PGMA-PEI [39,40].

### 3.4. Thermogravimetric Analysis

The TGA curves of the samples can be observed in Figure 4. The TGA profile of the PMMA shows two degradation steps: The first stage showed stable behavior below 285 °C and a weight loss of 5%, which was attributable to the evaporation of water and the breakdown of small molecules from the PMMA microspheres [38,41].

A sharp weight loss between 285 °C and 415 °C in the second stage was exhibited, rapidly losing weight until it reached a weight equilibrium at about 415 °C—the weight loss was about 90% at this stage.

Two similar stages were found in magnetic PMMA microspheres; they showed better stability after the introduction of Fe_3_O_4_, and the mass residue was higher than that of the pure PMMA microspheres.

The TGA profile of the magnetic PMMA-PGMA-PEI also exhibited two degradation steps; a weight loss of about 4% between 30 °C and 290 °C could be ascribed to the removal of absorbed water or small-molecule compounds. A sharp decrease in weight was observed between 290 °C and 420 °C—this may be assigned to the degradation of the PEI and PGMA chains. The thermal degradation profile was more complex due to the presence of new organic chains such as PEI.

### 3.5. Adsorption Behaviour of Magnetic PMMA-PGMA-PEI

#### 3.5.1. Effect of GMA Concentration

Different GMA concentrations were investigated to achieve the optimal adsorption efficiency. The effect of GMA concentration on GY was investigated at an absorbed dosage of 200 kGy (Figure 5). The GY increased to its maximum until the concentration of GMA was at 40 wt%. Furthermore, considering various factors, a concentration of 40 wt% GMA and a grafting rate of 206.18% were considered the best conditions for preparing magnetic PMMA-PGMA.

#### 3.5.2. Effect of Absorbed Dose

The absorbed dose is a key parameter in the grafting process that directly determines the number of active free radical centers and has an important influence on the grafting efficiency—in turn affecting the adsorption performance of the adsorbents [29,30]. As the yield of free radicals can be significantly affected by the absorbed doses, the effect of the absorbed dose on the adsorbing capacity was discussed. In this study, 0.2 g of magnetic PMMA-PGMA-PEI microspheres were weighed in a sample tube, which was prepared with different radiation doses from 20 to 120 kGy at a dose rate of 20 kGy/pass. Then, 20 mL of Ce(III) solution (200 mg/L) was added and oscillated in a constant-temperature oscillator at 298.15 K for 24 h until adsorption was saturated. The adsorbing capacity was measured and calculated.

As shown in Figure 6, the adsorbing capacity increased substantially with increases in the irradiation dose up to a dose of 80 kGy, at which point the adsorbing capacity reached its maximum. However, when the irradiation dose increased further, the adsorbing capacity decreased slightly.

In view of the above results, the highest adsorbing capacity was 146.17 mg/g when the absorbed doses was 80 kGy. Thus, 80 kGy was selected as the basic research object for the following investigations.

#### 3.5.3. Effect of pH

pH is regarded as one of the most important factors for the adsorption process of water-adsorbent interfaces as the pH influences the physicochemical behavior of an aqueous medium and especially its adsorption capacity. The effect of the pH on the percentage of adsorption of the Ce(III) solution at equilibrium was investigated in the range of 2.0–7.0 at 298.15 K. As observed from Figure 7, the uptake of Ce(III) was highly dependent on the pH value. Under strong acidic conditions, the sorption progressively increased with pHs between 2 and 3; this may be caused by the PEI groups being in the protonated form, and thus, the amount of adsorbed Ce(III) being very limited. As the pH increased from 3 to 6, the uptake capacity increased dramatically. This may be attributed to the functional PEI groups being deprotonated; thus, the number of electrostatic and ionic interactions increased. After the optimum pH, the amount of adsorbed Ce(III) no longer increased as the pH increased, which can be ascribed to the generation of metal precipitation and the formation of REE hydroxo colloids [16]. Combined with the above factors, pH = 6.0 was chosen as the optimum condition for the uptake of Ce(III).

#### 3.5.4. Effect of Dosage

This study also investigated the adsorbent doses’ effect on the Ce(III) adsorption capacity. Typically, magnetic PMMA-PGMA-PEI microspheres achieved a high adsorption efficiency, especially at the lower concentrations. According to Figure 8, when the magnetic PMMA-PGMA-PEI microsphere concentration increased within 0.25–2.5 g/L, the Ce(III) removal efficiency gradually increased from 24.96% to 94.75%. However, the adsorption volume gradually decreased from 201.77 to 76.43 mg/g because with the increase in the dosage of adsorbent resin, the number of active sites in the active increased, which enhanced the removal efficiency. However, excessive active sites were observed in the adsorbent at high dosages, which led to a decrease in the adsorption capacity for each adsorbent [29]. Considering both the adsorption capacity and removal efficiency, the 1.0 g/L magnetic PMMA-PGMA-PEI microspheres were considered to be the optimal dosage for Ce(III) adsorption.

#### 3.5.5. Initial Concentration and Adsorption Isotherm

This study analyzed how the initial concentration affected the adsorption, and the effects of various temperatures on the adsorption were also investigated. According to Figure 9A, as the Ce(III) initial concentration increased, the adsorption volume increased rapidly at first, and subsequently increased slowly. Finally, the adsorption equilibrium was achieved. The higher the temperature was, the more obvious this trend.

To investigate the relationship between the initial Ce(III) solution concentration and the adsorption capacity at various temperatures, the adsorption isotherms were studied. Adsorption isotherms were conducted at three temperatures (298.15, 308.15, 318.15 K) with diverse initial Ce(III) concentrations (50–450 mg·L^−1^). The mixed sample was shaken for 24 h to ensure adsorption equilibrium. Langmuir [37] as well as Freundlich models [35] were used to investigate the adsorption isotherms, as follows:(4)$$\frac{Ce}{Qe}=\frac{Ce}{Qm}+\frac{1}{K_{L}Q_{M}}$$
(5)$$\ln Q_{e}=\ln K_{F}+\frac{1}{n}\ln C_{e}$$
where *Q_e_* (mg·g^−1^) and *C_e_* (mg·L^−1^) represent the adsorption capacity and Ce(III) concentrations under equilibrium, respectively; *K_L_* (L/mg) is the Langmuir constant, *Q_m_* (mg/g) is the maximal adsorption capacity; 1/*n* represents the heterogeneity factor, and *K_F_* (mg/g) denotes the Freundlich constant.

The isotherm curves of the Ce(III) on the magnetic PMMA-PGMA-PEI are displayed in Figure 9B. Langmuir and Freundlich models were used to fit the adsorption isotherms (Figure 9C,D). Table 1 displays the obtained adsorption isotherm parameters. The Langmuir model exhibited an increased linear correlation (*R*^2^ > 0.99) compared with the Freundlich model (*R*^2^ < 0.84). Thus, the higher fitting degree of the Langmuir model revealed that Ce(III) adsorbed onto magnetic PMMA-PGMA-PEI was almost a monolayer process. In addition, the calculated maximum adsorption capacity was 186.57 mg/g at 318.15 K, which is very close to the experimental result of 189.81 mg/g.

#### 3.5.6. Contact Time and Adsorption Kinetics

As shown in Figure 10A, the adsorption capacity gradually increased with the prolongation of time, and the adsorption equilibrium was basically reached at 360 min at 298.15 K.

For further exploring the adsorption behavior and rate-control steps, two kinetic adsorption models were investigated in Figure 10B,C. Samples were obtained at specific intervals and tested for remaining Ce(III) concentrations. The adsorption kinetics of the Ce(III) on the magnetic PMMA-PGMA-PEI were fitted with pseudo-first order [26] and pseudo-second order models [29], respectively. The two models are presented in the flowing equations:(6)$$\frac{dQ_{t}}{dt}=k_{1}(Q_{e}-Q_{t})$$
(7)$$\frac{dQ_{t}}{dt}=k_{2}{\left(Q_{e}-Q_{t}\right)}^{2}$$
where *Q**_t_* and *Q**_e_* (mg/g^−1^) stand for the Ce(III) adsorption capacity at time *t* and equilibrium, separately; *k*_1_ (min^−1^) and *k*_2_ (g·mg^−1^·min^−1^) represent the equilibrium constants for pseudo-first order as well as pseudo-second order models, separately.

Obviously, the *R*^2^ of the pseudo-second order model from Table 2 was larger than that of the pseudo-first order model. Similarly, the *Q*_e,cal_ calculated by the pseudo-second order model was more consistent with the adsorption capacity than that of pseudo-first order model. Based on the aforementioned findings, chemisorption controlled the absorption of Ce(III) on magnetic PMMA-PGMA-PEI.

#### 3.5.7. Adsorption Thermodynamics

To explore the effect of temperature on the Ce(III) uptake process, an uptake experiment was conducted under different temperatures (298.15 K, 308.15 K, 318.15 K). These thermodynamic parameters and values—entropy change Δ*S* (J·mol^−1^ K^−1^), enthalpy change Δ*H* (kJ·mol^−1^) and Gibbs free energy change Δ*G* (kJ·mol^−1^)—were obtained according to the following equations [37]:(8)$$\ln(\frac{Qe}{Ce})=-\frac{\Delta H}{RT}\times \frac{1}{T}+\frac{\Delta S}{R}$$
(9)ΔG=ΔH−TΔS

All the thermodynamic parameters and values for Ce(III) adsorption onto the magnetic PMMA-PGMA-PEI for various concentrations were calculated and are presented in Table 3. As showed in Table 3, the calculated negative values of Δ*G* implied the spontaneous and thermodynamically favorable process of Ce(III) adsorption onto magnetic PMMA-PGMA-PEI. Furthermore, the values of Δ*G* decreased as the temperature increased from 298.15 K to 318.15 K, which indicated that increases in temperature were conducive to the spontaneous progress of Ce(III) adsorption. According to the results, an increased temperature led to a higher adsorption capacity, which indicated that Ce(III) adsorption belonged to an endothermic reaction. With increases in the initial concentration of Ce(III) from 150 to 300 mg/L, the adsorption process became slower—which was ascribed to the sorption sites being occupied.

Furthermore, the positive Δ*S* indicated that the adsorption of Ce(III) was random on the surface of the magnetic PMMA-PGMA-PEI. In addition, the adsorption process increased the freedom of the entire system. Here, Δ*H* exhibited a positive value—thus representing that the adsorption process was endothermic, which occurs more easily at a higher temperature. This result is associated with the increased active sites at high temperatures [42].

### 3.6. Comparative Analysis of Ce(III) Adsorption

Table 4 shows a comparison of the maximum uptake capacities (Qmax) of magnetic PMMA-PGMA-PEI for Ce(III) with other adsorbents. It can be seen that the maximum adsorption capacity of magnetic PMMA-PGMA-PEI was superior to most of the adsorbents proposed in previous works. Furthermore, the irradiation grafting preparation method was easy to operate and environmentally friendly. The excellent uptake capacity could be mainly attributed to the sufficient exposure of the active sites and the high surface reactivity.

## 4. Conclusions

Here, a new adsorbent PEI functionalized magnetic PMMA-PGMA was successfully prepared via the electron beam irradiation grafting method followed by chemical modification, which was affirmed by FT-IR, XRD, SEM and TG. The as-prepared magnetic PMMA-PGMA-PEI microspheres were used to enhance the high-efficiency uptake of Ce(III) from aqueous solutions. The obtained magnetic microspheres had good magnetism, satisfactory thermostability and well-defined morphology. The optimum uptake pH was 6.0, and the maximum adsorbing capacity of the Ce(III) was 189.81 mg/g at 318.15 K in the system with the adsorbent dosage of 1.0 g/L. In addition, the Ce(III) uptake could be accurately described by the Langmuir isotherm model and pseudo-second order kinetic model. According to the thermodynamic parameters, the Ce(III) adsorption was an entropy-driven endothermic process. Moreover, the Ce(III) uptake was mainly achieved by chelation interactions between lone-pair electron donations of amines in PEI and the Ce(III). The Van der Waals interactions and ion exchange were also greatly promoted the adsorption. Thus, the as-prepared magnetic PMMA-PGMA-PEI microspheres have great potential in the field of Ce(III) removal.

## Data Availability

Not applicable.

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
