# Peer review of "Green Preparation of Aminated Magnetic PMMA Microspheres via EB Irradiation and Its Highly Efficient Uptake of Ce(III)"

_materials, 2022, doi:10.3390/ma15196553_

Round 1

Reviewer 1 Report

Dear Editor,

It is a good paper. The research article presents an affordable approach for the sequestration of Ce(III) by Magnetic PMMA-PGMA-PEI adsorbent. The present research article has been presented as per journal standards, though the claim to get published in the journal need some revisions. This manuscript can be accepted for publication if author addresses the comments listed below. The topic of the manuscript is interesting. However, after careful reading the text, the authors should consider the following issues while preparing the major revised manuscript:

1.      The main objective of the work must be written on the more clear and more concise way at the end of introduction section.

2.      The novelty of the work must be clearly addressed and discussed, compare your research with existing research findings and highlight novelty.

3.      Why Ce(III) was chosen in this study?

4.      How the authors confirmed that the removal of Ce(III) was through adsorption only not by another mechanisms such as ion exchange?

5.      Did you check the stability of the Ce(III) in the function of pH? Any change of structure like an example?

6.      The manuscript is readable. I do understand what the authors mean, but Quality of English language needs minor revision. Check the punctuation marks, grammar and spelling errors in the research article thoroughly. Authors need get it checked by someone who is proficient in English language or use any grammar related software instead.

7.      Please add more quantitative results in the abstract.

8.      The abstract is concise and accurately summarizes the essential information. Abstract should be rewritten to summarize the work; the abstract should briefly state the purpose of the research, the principal results, and major conclusions. An abstract is often presented separately from the article, so it must be able to stand alone.

9.      The logic of the current introduction should be revised, and I suggest organizing the Introduction section as following order: importance and meanings, previous studies (literature review), the gaps of previous studies, and objectives of this study.

10.  The innovation of this study is not well described in "INTRODUCTION" section. Recheck it. The "Introduction" section needs a minor rewrite. For example, the application of adsorption pathway for wastewater treatment should be adequately reviewed. Use the following references that are relevant and up to date to improve the quality of this section.

* A critical review of biosorption of dyes, heavy metals and metalloids from wastewater as an efficient and green process.

* Insight on water remediation application using magnetic nanomaterials and biosorbents.

* Biosorption, an efficient method for removing heavy metals from industrial effluents: A Review.

* Biosorption of heavy metals from aqueous solution by various chemically modified agricultural wastes: A review

11. The discussion of thermodynamics parameters needs improvement.

12. Captions of the figure and tables must be with complete information, conditions etc

13. Did the authors use the linear form of isothermal equations or the nonlinear form?

14. How did the authors achieve equilibrium time in the experiments?

‎15. Please improve the conclusion with clear quantitative findings.

Author Response

Dear reviewers,

Thanks very much for your time and efforts spent in editing my manuscript (Manuscript ID: materials-1866089). Your suggestions and queries have greatly improved the quality of this manuscript. We have revised our manuscript carefully following your comments and highlighted all alteration to the original submission in RED.

We sincerely hope that our revision have met the expectation of the editor and reviewers. Please don’t hesitate to let us know if the manuscript is found to contain any mistakes.

Author Response

Dear reviewer,

Thanks very much for your time and efforts spent in editing my manuscript (Manuscript ID: materials-1866089). Your suggestions and queries have greatly improved the quality of this manuscript. We have revised our manuscript carefully following your comments and highlighted all alteration to the original submission in RED.

We sincerely hope that our revision have met the expectation of the editor and reviewers. Please don’t hesitate to let us know if the manuscript is found to contain any mistakes.

Round 2

Reviewer 2 Report

I have go through the revised manuscript and found that authors have significantly improved the quality of the manuscript. However, BET and XPS investigations were asked to incorporate in the revised version. But, the authors ignored the suggestion. Why? 

Author Response

Response to Reviewers' Comments

Dear reviewer,

Thanks very much for your time and efforts spent in editing my manuscript (Manuscript ID: materials-1866089). Your suggestions and queries have greatly improved the quality of this manuscript.

We sincerely hope that our revision have met the expectation of the editor and reviewers. Please don’t hesitate to let us know if the manuscript is found to contain any mistakes.

Reviewer Comments:I have go through the revised manuscript and found that authors have significantly improved the quality of the manuscript. However, BET and XPS investigations were asked to incorporate in the revised version. But, the authors ignored the suggestion. Why? 

Answer: Thanks a lot for your kind and careful review. We have supplemented the BET testing. The N2 adsorption-desorption isotherms of Magnetic PMMA and Magnetic PMMA-PGMA-PEI was shown in Fig.1 and the BET testing parameters were list in Table 1.The adsorption isotherm of Magnetic PMMA and Magnetic PMMA-PGMA-PEI both are type â…£, and the hysteresis loop of adsorption isotherm is type H4, which has a clear micropore filling phenomenon at low pressure,The generation of type H4 hysteresis loop shows that a large number of mesopores appear in Magnetic PMMA and Magnetic PMMA-PGMA-PEI[1].The Brunauer-Emmett-Teller (BET) surface area, Barrett-Joyner-Halenda (BJH) pore volume and average pore diameter of Magnetic PMMAwere 38.5259 m2·g1, 0.3145 cm3·g−1 and 3.4497 nm, respectively, according to N2 adsorption/desorption isotherms. The BET surface area , pore volume and average pore diameter of Magnetic PMMA-PGMA-PEI were lower those of pristine Magnetic PMMA(25.2800 m2·g−1, 0.1843 cm3·g−1 and 1.7236 nm) because the grafting of organic functional groups occupied some spaces of the pores of Magnetic PMMA after chemical modifification, resulting in reduced specifific surface area, pore volume and average pore diameter of Magnetic PMMA-PGMA-PEI[2].

Fig.1 N2 adsorption-desorption isotherms of Magnetic PMMA and Magnetic PMMA-PGMA-PEI

Table 1.BET testing parameters of Magnetic PMMA and Magnetic PMMA-PGMA-PEI

Surface Area(m2/g)

Pore Volume(mL/g)

Pore diameter Dv(d)(nm)

Magnetic PMMA

38.5259

0.3145

3.4497

Magnetic PMMA-PGMA-PEI

25.2800

0.1843

1.7236

[1] X. Ma, S. Zhao, Z. Tian, G. Duan, H. Pan, Y. Yue, S. Li, S. Jian, W. Yang, K. Liu, S. He, S. Jiang, MOFs meet wood: reusable magnetic hydrophilic composites toward efficient water treatment with super-high dye adsorption capacity at high dye concentration,Chemical Engineering Journal,2022,446:136851.

[2]Zhang Y N ,  Guo J Z ,  Wu C , Chen L, Li B. Enhanced removal of Cr(VI) by cation functionalized bamboo hydrochar[J]. Bioresource Technology, 2022, 347:126703-126709.     

   We are very sorry that we have not supplemented the XPS test, because it does not have XPS test equipment in our university, and it will take at least 15 days to send sample to other units for testing. However, as the editor request us to upload the revised file within 3 days. Thus the characterization of XPS can not be conducted in such a short time. In our subsequent research work, we will adopt the comments and suggestions of reviewers to add the XPS characterization to further improve the quality of our works. Special thanks to you for your good comments.

We appreciate for Editors/Reviewers’ warm work earnestly, and hope that the correction will meet with approval.Once again, thanks very much for your comments and suggestions

Sincerely,

                                                          Tao Chen
